# Graph Neural Flows for Unveiling Systemic Interactions Among Irregularly Sampled Time Series

**Giangiacomo Mercatali** *
HES-SO Genève
University of Manchester
giangiacomo.mercatali@hesge.ch

**Andre Freitas**
Idiap Research Institute
University of Manchester
NBC, CRUK Manchester Institute
andre.freitas@idiap.ch

**Jie Chen**
MIT-IBM Watson AI Lab
IBM Research
chenjie@us.ibm.com

## Abstract

Interacting systems are prevalent in nature. It is challenging to accurately predict the dynamics of the system if its constituent components are analyzed independently. We develop a graph-based model that unveils the systemic interactions of time series observed at irregular time points, by using a directed acyclic graph to model the conditional dependencies (a form of causal notation) of the system components and learning this graph in tandem with a continuous-time model that parameterizes the solution curves of ordinary differential equations (ODEs). Our technique, a graph neural flow, leads to substantial enhancements over non-graph-based methods, as well as graph-based methods without the modeling of conditional dependencies. We validate our approach on several tasks, including time series classification and forecasting, to demonstrate its efficacy.

## 1 Introduction

Real-life dynamical systems consist of a group of components interacting in a complex manner. With time series data for each component, predicting the system dynamics remains challenging, because modeling each component independently is straightforward while accounting for their interactions is hard without a priori knowledge. Sometimes, these interactions are causal. For example, "phantom jams" in which a small disturbance (e.g., a driver hitting the brake too hard) in a heavy traffic can be amplified over a large area of the transportation network [16]. While traffic congestion often enjoys spatial proximity, outage of a power network can be propagated non-locally over the grid; i.e., a sequence of blackouts jumps across hundreds of kilometers [21]. We pose this question: What time series models best capture the interactive nature of different system dynamics?

Graph neural networks (GNNs) [49, 46] are modern tools to enhance time series models when multiple time series are interconnected by a given graph [33]. In these models, time series are encoded by using a recurrent neural network (RNN); at every time step, a GNN is used to aggregate features over the graph. When the graph is unknown, graph-structure learning approaches have been proposed; some approaches learn a single interaction graph over time [30, 41, 11] while others infer a different one at each time step [20]. These models are in general discrete-time models, suitable for regularly spaced time points [39, 40].

---

*Work done while at the University of Manchester

38th Conference on Neural Information Processing Systems (NeurIPS 2024).

To handle irregular time points, we consider continuous-time models. The celebrated neural ordinary differential equation (ODE) technique [7] models a time series as the solution of an unknown ODE and optimizes the ODE parameters by using gradients computed through the adjoint. Neural ODEs were later adopted for latent variable modeling [39, 5], which introduced discontinuities at observed time points for reducing prediction errors and variances. Neural ODEs were also adopted for graph-based modeling [38, 24, 25, 26, 10, 27, 3], where the equation accounts for multiple time series and the right-hand side uses a graph to associate the different series. These models construct a graph, which could be time-dependent, in various manners, such as based on node features, co-observations within a sliding window, latent representations, or attentions.

In this work, we propose *learning* a graph that reveals the dependency structure of the time series. To this end, we consider a form of causal notation—the Bayesian network [36, 37]—which is a directed acyclic graph (DAG), where a node is conditionally independent of its non-descendents given its parents [34, 12, 43, 22]. Such a conditional dependence structure specifies how component dynamics depends on each other. This model has a potential for causal discovery when one interprets the learned graph unknown a priori (e.g., how blackouts cascade over the power grid, whose known topology differs from the unknown influence graph [21]). More importantly, when modeled properly, the graph can improve the performance of downstream tasks because of the capturing of systemic interactions.

Our proposed model is a *graph neural flow* (GNeuralFlow). A neural flow [4] is the learned solution of an unknown ODE based on irregularly sampled time series; it is advantageous over the neural ODE technique in that it models directly the ODE solution rather than the right-hand side, thus avoiding repeating calls of a numerical solver, whose cost could be expensive. We condition multiple neural flows, one for each time series, on the DAG, and we instantiate their interactions as a GNN; e.g., a graph convolutional network, GCN [31]. The graph convolution therein augments the parameterization of the ODE solution by aggregating the information of the neighboring time series at each time point, fitting a graph-conditioned ODE that models the interacting system.

Thus, GNeuralFlow is advantageous over prior graph ODE approaches (e.g., GDE [38], LG-ODE [24], CG-ODE [25], CF-GODE [26], STG-NCDE [10], MTGODE [27], RiTINI [3]) in two aspects. First, through learning, the graph reveals the conditional dependencies of the time series, offering a more intuitive structure for analysis. Second, it removes the reliance on numerical ODE solvers and gains computational efficiency. We demonstrate empirical evidence to show that GNeuralFlow outperforms graph ODE approaches in several downstream tasks, on both synthetic and real-life data.

We highlight the following contributions of this work:

- We propose a novel graph-based continuous-time model GNeuralFlow for learning systemic interactions. The interactions are modeled as a Bayesian network, which can be learned in tandem with other model parameters.

- We design model parameterizations by leveraging GNNs to encode the systemic interactions. These parameterizations can additionally be used in latent variable modeling.

- We demonstrate the use of GNeuralFlow in regression problems and latent variable modeling and show the performance improvement in several time series classification and forecasting benchmarks.

## 2   Background: Neural ODE and Neural Flows

Denote by $\mathbf{x}(t) \in \mathbb{R}^d$ the solution of an ODE

$$\dot{\mathbf{x}} = f(t, \mathbf{x}) \tag{1}$$

under well-behaving conditions (e.g., a specified initial condition of $\mathbf{x}$ and Lipschitz continuity of $f$). *Neural ODE* [7] is a modeling technique that allows uncovering the trajectory $\mathbf{x}(t)$ without a known right-hand side $f$. The technique parameterizes $f$ by a neural network with parameters $\boldsymbol{\theta}$ such that the trajectory $\mathbf{x}$ is a function of $\boldsymbol{\theta}$. Through matching the trajectory with observed data at a few (possibly irregular) time points by using a loss function $L$, the vector field $f$ is unveiled. Given the initial condition $\mathbf{x}(t_0) = \mathbf{x}_0$, we write $\mathbf{x}(t_0), \ldots, \mathbf{x}(t_N) = \text{ODESolve}(f, \mathbf{x}_0, (t_0, \ldots, t_N))$, where the solutions at times $t_1, \ldots, t_N$ are obtained by invoking any blackbox numerical ODE solver (such as Runge–Kutta [17]). The training of the model parameters $\boldsymbol{\theta}$ requires the gradient $\nabla_{\boldsymbol{\theta}} L$, which can

be economically computed by using the adjoint $\mathbf{a} := \nabla_{\mathbf{x}} L$ rather than expensively back-propagating through the ODE solver.

Neural ODE has two far-reaching impacts. First, it is a continuous-time technique, which is a better alternative to discrete-time techniques (such as RNNs) for modeling irregularly sampled time series [39]. Second, it leads to a continuous version of the *normalizing flow* [35].

Other than directly modeling the observed data $\mathbf{x}$, Chen et al. [7] proposed to use neural ODEs as latent variable models, which model the latents $\mathbf{z}$ instead; that is, $\dot{\mathbf{z}} = f(t, \mathbf{z})$. A straightforward idea is to build a variational autoencoder (VAE) [29], where the encoder is an RNN that evolves the hidden state $\mathbf{h}(t)$ over training time points and concludes a latent variable $\mathbf{z}_0$ as the ODE initial condition. A drawback is that this approach still uses a discrete-time model (RNN) to handle irregularly sampled observations. Two approaches mitigating this drawback are ODE-RNN [39] and GRU-ODE-Bayes [5]. Both approaches demonstrate smaller prediction errors and variances compared with the vanilla neural ODE + VAE approach.

Another drawback of neural ODE is that it invokes a numerical solver, often multiple times in the adjoint computation because of multiple time intervals, which can be rather time consuming. A *neural flow* [4] is an alternative to neural ODE as it models the solution of (1) directly:

$$\mathbf{x}(t) = F(t, \mathbf{x}_0),$$

by using a parameterized function $F$ that depends on the initial condition $\mathbf{x}_0$. Optimizing the parameters of $F$ can be more efficient because ODE solvers are no longer needed. A neural flow is not to be confused with a normalizing flow. Neural flows can replace the use of neural ODEs in latent variable models ODE-RNN and GRU-ODE-Bayes.

## 3 DAG-Based ODE for Modeling Systemic Interactions

In this section, we motivate the form of ODE considered in this paper based on DAG modeling.

### 3.1 DAG Model for Systemic Interactions

In probabilistic graphical models, the conditional dependence structure is a principled framework for modeling systemic interactions. Therein, a *Bayesian network* [36] of $n$ random variables $y^1, \ldots, y^n$ is a DAG with these variables as the nodes. Let $\mathbf{A} \in \mathbb{R}^{n \times n}$ be the (weighted) adjacency matrix of the DAG, where $a_{ij} \neq 0$ means that $y^i$ is a parent of $y^j$. A Bayesian network describes the conditional dependence structure of the variables; namely, a node is conditionally independent of its non-descendents given its parents. Therefore, the joint probability $p(y^1, \ldots, y^n)$ can be factorized into a much simpler form: $p(y^1, \ldots, y^n) = \prod_{j=1}^{n} p(y^j \mid \mathrm{pa}(y^j))$, where $\mathrm{pa}(y^j) = \{y^i : a_{ij} \neq 0\}$ denotes the parent set of $y^j$. The conditional dependence is a necessary condition for the causal relationship between parent $y^i$ and child $y^j$ [37].

The (linear) structural equation model, SEM [45, 14], is a commonly used tool to further quantify the conditional probabilities. Without loss of generality, assume that the random variables are topologically sorted according to the partial ordering $\prec$, where $i \prec j$ iff there exists an edge from $i$ to $j$. Then, the DAG adjacency matrix $\mathbf{A} = [a_{ij}]$ is strictly upper triangular. The SEM model reads

$$
\begin{aligned}
y^1 &= &\epsilon_1 \\
y^2 &= a_{12}y^1 &+ \epsilon_2 \\
y^3 &= a_{13}y^1 + a_{23}y^2 &+ \epsilon_3 \\
&\vdots \\
y^n &= a_{1n}y^1 + a_{2n}y^2 + \cdots + a_{n-1,n}y^{n-1} + \epsilon_n,
\end{aligned}
$$

where the residuals $\epsilon_1, \ldots, \epsilon_n$ are (possibly correlated) Gaussian noises. In this model, $y^j$ depends on $y^i$ only when $i < j$. Importantly, some of the above $a_{ij}$'s can be zero. Then, $y^j$ is independent of such $y^i$'s given the rest.

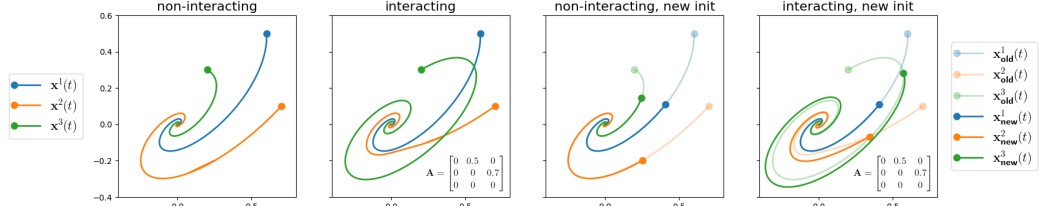

Figure 1: Left two: Trajectories of a non-interacting system and an interacting system (using interaction matrix $\mathbf{A}$), under the same initial conditions. Right two: Replica of the left two systems but the initial conditions are changed. Trajectories change on the rightmost plot.

## 3.2   DAG-Based ODE as Continuous-Time Models

Consider an autonomous ODE $\dot{\mathbf{x}} = \mathbf{B}\mathbf{x}$ with the initial condition $\mathbf{x}(0) = \mathbf{x}_0 \in \mathbb{R}^2$. This ODE describes a 2D vector field $\mathbf{B}\mathbf{x}$; each solution curve $\mathbf{x}(t) = \mathrm{expm}(\mathbf{B}t)\mathbf{x}_0$ given the initial point $\mathbf{x}_0$ is a streamline instantaneously tangential to the vector field. Throughout the paper, we use $\mathrm{expm}$ to denote matrix exponential for matrix arguments and $\exp$ to denote element-wise exponential.

Now consider $n$ trajectories $\mathbf{x}^1(t), \ldots, \mathbf{x}^n(t)$. Inspired by SEM, we model that (i) the vector fields that generate the $n$ trajectories follow the same conditional dependence structure governed by $\mathbf{A}$ and (ii) the residual $\mathbf{B}\mathbf{x}^j - \sum_{i=1}^{j-1} a_{ij}\mathbf{B}\mathbf{x}^i$ gives the velocity $\dot{\mathbf{x}}^j$ for each $j$. Mathematically,

$$
\begin{aligned}
\mathbf{B}\mathbf{x}^1 &= & & \dot{\mathbf{x}}^1 \\
\mathbf{B}\mathbf{x}^2 &= a_{12}\mathbf{B}\mathbf{x}^1 & & + \dot{\mathbf{x}}^2 \\
\mathbf{B}\mathbf{x}^3 &= a_{13}\mathbf{B}\mathbf{x}^1 + a_{23}\mathbf{B}\mathbf{x}^2 & & + \dot{\mathbf{x}}^3 \\
&\ \ \vdots \\
\mathbf{B}\mathbf{x}^n &= a_{1n}\mathbf{B}\mathbf{x}^1 + a_{2n}\mathbf{B}\mathbf{x}^2 + \cdots + a_{n-1,n}\mathbf{B}\mathbf{x}^{n-1} + \dot{\mathbf{x}}^n.
\end{aligned}
\tag{2}
$$

In other words, the solution curve $\mathbf{x}^j$ for each $j$ and any initial point $\mathbf{x}_0^j$ is a streamline instantaneously tangential to the residual field.

The behaviors of non-interacting and interacting systems are fundamentally different. Figure 1 illustrates an example. The first plot shows three independent trajectories satisfying $\dot{\mathbf{x}}^i = \mathbf{B}\mathbf{x}^i$ with initial conditions $\mathbf{x}_0^i$ for $i = 1, 2, 3$. The second plot shows three conditionally dependent trajectories, under the same initial conditions, but interacting through the DAG adjacency matrix

$$
\mathbf{A} = \begin{bmatrix} 0 & 0.5 & 0 \\ 0 & 0 & 0.7 \\ 0 & 0 & 0 \end{bmatrix}.
$$

Because $\mathbf{x}^1$ is independent of the rest, it is the same in both plots; but in the second plot, because $\mathbf{x}^2$ depends on $\mathbf{x}^1$ and $\mathbf{x}^3$ depends on $\mathbf{x}^2$, these two trajectories are different from their counterparts in the first plot. Moreover, in the third and fourth plots, we change the initial conditions. As long as the new initial points are along the original trajectories, the new trajectories still follow the old ones when $\mathbf{A}$ does not exist; however, when $\mathbf{A}$ exists, $\mathbf{x}^2$ and $\mathbf{x}^3$ deviate from the original trajectories, because of the conditional dependence.

## 3.3   From Linear Dependence to General

In general, the dependence among the time series may not be linear, even though structurally it is governed by the matrix $\mathbf{A}$. For example, a nonlinear SEM may inspire the ODE system $\mathbf{B}\mathbf{x}^1 = \dot{\mathbf{x}}^1$, $\mathbf{B}\mathbf{x}^2 = \mathbf{B}\mathbf{x}^1 + \dot{\mathbf{x}}^2$, $\mathbf{B}\mathbf{x}^3 = \mathbf{B}\mathbf{x}^1 \odot \mathbf{B}\mathbf{x}^2 + \dot{\mathbf{x}}^3$, where the dependence of $\mathbf{x}^3$ on $\mathbf{x}^1$ and $\mathbf{x}^2$ is not linear. Thus, we consider general ODE systems of the form

$$
\dot{\mathbf{x}}^j = f(t, \{\mathbf{x}^j\} \cup \mathrm{pa}(\mathbf{x}^j)), \quad j = 1, \ldots, n,
\tag{3}
$$

where recall that $\mathrm{pa}(\mathbf{x}^j)$ denotes the set of parents of $\mathbf{x}^j$. Equivalently, we write $\dot{\mathbf{X}} = f(t, \mathbf{X}, \mathbf{A})$ in the matrix form. Note that (2) is a special case of (3), which can be written as $\dot{\mathbf{X}} = (\mathbf{I} - \mathbf{A}^\top)\mathbf{X}\mathbf{B}^\top$. Note also that (3) is permutation equivariant and $\mathbf{A}$ can be any DAG matrix, not necessarily upper triangular ones.

# 4 GNeuralFlow: An ODE-Solver-Free Method

## 4.1 Problem Setup and Model Framework

**Problem 1.** Let $\mathbf{A} \in \mathbb{R}^{n \times n}$ be the weighted adjacency matrix of a DAG and let $\mathbf{X}(t) : \mathbb{R} \to \mathbb{R}^{n \times d}$ be the solution curve of the initial-value ODE system

$$\dot{\mathbf{X}} = f(t, \mathbf{X}, \mathbf{A}) \quad \text{with} \quad \mathbf{X}(0) = \mathbf{X}_0, \tag{4}$$

where the right-hand side $f$ is unknown.[2] Given data $\mathbf{X}(t_0), \ldots, \mathbf{X}(t_N)$ at irregular time points, develop a model that predicts $\mathbf{X}(t)$ for any $t \geq t_0$ as well as $\mathbf{A}$. To account for practical use, at some time point $t_j$, some rows of $\mathbf{X}(t_j)$ may be missing.

A growing body of research addresses the problem when $\mathbf{X}$ has a single row (i.e., $n = 1$; hence, $\mathbf{A}$ is irrelevant), notably through using a neural network to parameterize $f$ and using a numerical solver to evaluate $\mathbf{X}$ at $t_0, \ldots, t_N$ [7, 39, 5]. When $n > 1$ and the rows of the system are independent (i.e., $f$ is identically the same function for each row), these methods straightforwardly apply through batch training. However, when the rows of the system are not independent, the problem becomes rather challenging. One may flatten (4) into an $nd$-dimensional problem, but such a high dimension renders approaches using numerical solvers too costly.

Instead, we use a neural network to parameterize the solution of (4) directly; i.e.,

$$\mathbf{X}(t) = F(t, \mathbf{X}_0, \mathbf{A}), \tag{5}$$

where the solution $F$ is a function of $t$ but depends on the initial $\mathbf{X}_0$ as well as $\mathbf{A}$. The neural network parameterization cannot be entirely free. First, the solution $F$ should satisfy the initial condition $F(0, \mathbf{X}_0, \mathbf{A}) = \mathbf{X}_0$. Second, the fundamental theorem on flows [32, Theorem 9.12] asserts that every smooth $f$ with an initial condition determines a unique $F(t, \mathbf{X}_0, \mathbf{A})$ and for any $t$, $F(t, \cdot, \mathbf{A})$ is a diffeomorphism. Therefore, we formulate our model for Problem 1 in the following.

**Solution framework.** The model for $\mathbf{X}(t)$ is a neural network $F(t, \mathbf{X}, \mathbf{A})$ that satisfies:

1. $F(0, \mathbf{X}_0, \mathbf{A}) = \mathbf{X}_0$;

2. $F(t, \mathbf{X}, \mathbf{A})$ is invertible in $\mathbf{X}$ for any $t$ and $\mathbf{A}$; equivalently, the streamline $F(t, \mathbf{X}_0, \mathbf{A})$ given any $\mathbf{X}_0$ and $\mathbf{A}$ is not self-intersecting.

## 4.2 Graph Encoder

We will make heavy use of a GNN to encode the DAG adjacency matrix $\mathbf{A}$ for parameterizing $F$. For simplicity, we employ the seminal architecture GCN. A (popularly used) two-layer GCN reads

$$\widetilde{\mathbf{X}} = \text{GCN}(\mathbf{A}, \mathbf{X}) = \widehat{\mathbf{A}} \, \text{ReLU}(\widehat{\mathbf{A}} \mathbf{X} \mathbf{W}) \mathbf{U}, \tag{6}$$

where $\mathbf{X}$ is the input node feature matrix, $\widetilde{\mathbf{X}}$ contains the transformed features, and $\mathbf{W}$ and $\mathbf{U}$ are parameters. GCN defines $\widehat{\mathbf{A}}$ as a symmetric normalization of $\mathbf{A}$, but many alternatives are viable, such as a simple scaling of $\mathbf{A}$. We also consider

$$\widehat{\mathbf{A}} = \mathbf{I} - \mathbf{A}^\top / \gamma, \quad \text{where} \quad \gamma = \max_j \left\{ \sum_{i \neq j} |\mathbf{B}_{ij}| \right\} \text{ and } \mathbf{B} = \mathbf{A} + \mathbf{A}^\top. \tag{7}$$

This definition is motivated by SEM, where $\mathbf{I} - \mathbf{A}^\top$ is the operator (Section 3.3). Here, $\mathbf{A}$ is not symmetrized because doing so cannot distinguish edge directions. A benefit of taking (7) is that the scaling factor $\gamma$ leads to a bounded spectral norm, which is an ingredient of invertibility required by the **Solution framework**.

**Theorem 1.** For any DAG adjacency matrix $\mathbf{A}$, the matrix $\widehat{\mathbf{A}}$ defined in (7) admits $\|\widehat{\mathbf{A}}\|_2 \leq 2$.

---

[2] In practice, the initial time point $t_0$ may not be zero, in which case one may shift the ODE along the temporal dimension by $t_0$.

### 4.3 Parameterization of $F$

The neural network $F$ in (5) can be defined in several ways by incorporating the graph encoder while satisfying the **Solution framework**.

**ResNet flow.** The first design is the ResNet architecture

$$F(t, \mathbf{X}, \mathbf{A}) = \mathbf{X} + \varphi(t) \cdot g(t, \mathbf{X}, \mathbf{A}), \tag{8}$$

which is a building block of invertible networks [1]. Here, $\varphi(t)$ satisfies $\varphi(0) = 0$ such that the requirement $F(0, \mathbf{X}_0, \mathbf{A}) = \mathbf{X}_0$ is met. Additionally, if $\varphi(\cdot) \in [0, 1]$ and $g(t, \cdot, \mathbf{A})$ is a contractive mapping, then $F(t, \cdot, \mathbf{A})$ is invertible.

We let $\varphi$ be the tanh function and parameterize $g$ by using two MLPs together with a GCN:

$$g(t, \mathbf{X}, \mathbf{A}) = \mathrm{MLP}^1(\mathbf{X}||\widetilde{\mathbf{X}}||t) \odot \mathrm{MLP}^2(\mathbf{X}||t), \qquad \widetilde{\mathbf{X}} = \mathrm{GCN}(\mathbf{A}, \mathbf{X}), \tag{9}$$

where $||$ denotes concatenation row-wise and each MLP acts on the input matrix row-wise independently. The neural network $g$ is generally not contractive, but bounding the spectral norm of each linear layer can theoretically guarantee contraction of an MLP [19]. Moreover, Theorem 1 indicates that bounding the spectral norm of the GCN parameters can guarantee contraction of GCN as well (because $\|\widehat{\mathbf{A}}\|_2$ is bounded). Thus, in theory, $g$ can be made contractive. In practice, regularization is used to encourage a small Lipschitz constant of $g$ [19].

**GRU flow.** The second design mimics the GRU [9]:

$$F(t, \mathbf{X}, \mathbf{A}) = \mathbf{X} + \varphi(t) \cdot h^1(t, \mathbf{X}) \odot h^2(t, \widetilde{\mathbf{X}}), \qquad \widetilde{\mathbf{X}} = \mathrm{GCN}(\mathbf{A}, \mathbf{X}), \tag{10}$$

where $h^k$, $k = 1, 2$, is computed by

$$r^k(t, \mathbf{X}) = \beta \cdot \mathrm{sigmoid}(f_r^k(t, \mathbf{X})), \qquad c^k(t, \mathbf{X}) = \tanh(f_c^k(t, r^k(t, \mathbf{X}) \odot \mathbf{X})),$$
$$z^k(t, \mathbf{X}) = \alpha \cdot \mathrm{sigmoid}(f_z^k(t, \mathbf{X})), \qquad h^k(t, \mathbf{X}) = z^k(t, \mathbf{X}) \odot (c^k(t, \mathbf{X}) - \mathbf{X}).$$

The base form $\mathbf{X} + \varphi(t) \cdot h(t, \mathbf{X})$, analogous to ResNet, comes from [5], who derived an ODE with the right-hand side being $h$ through algebraic manipulation of the GRU. We extend the base form by including an analogous term $h^2$ that incorporates the graph encoder. It can be shown that $F$ is invertible under a deliberate choice of $\alpha$ and $\beta$ when the MLPs $f_z^k$, $f_r^k$, $f_c^k$ and the GCN are contractive. As discussed earlier, Theorem 1 indicates that GCN can be made contractive similarly as the MLPs through bounding the spectral norm of their parameters.

**Theorem 2.** If $f_z^k(t, \cdot)$, $f_r^k(t, \cdot)$, $f_c^k(t, \cdot)$, and $\mathrm{GCN}(\mathbf{A}, \cdot)$ are contractive, the function $F(t, \cdot, \mathbf{A})$ defined in (10) is invertible whenever $\alpha(5\beta + 6) \leq 2$.

**Coupling flow.** For the third design, we use normalizing flows, because they are invertible by definition. An example of the normalizing flow is the coupling flow [13]. Let $\{U, V\}$ be a partitioning of the column indices $1, \ldots, d$. With the graph encoder, we extend a usual coupling flow block to the following:

$$F(t, \mathbf{X}, \mathbf{A})_U = \mathbf{X}_U \odot \exp\left(\varphi_u(t) \cdot u(t, \mathbf{X}_V, \widetilde{\mathbf{X}}_V)\right) + \left(\varphi_v(t) \cdot v(t, \mathbf{X}_V, \widetilde{\mathbf{X}}_V)\right)$$
$$F(t, \mathbf{X}, \mathbf{A})_V = \mathbf{X}_V, \qquad\qquad\qquad \widetilde{\mathbf{X}}_V = \mathrm{GCN}(\mathbf{A}, \mathbf{X}_V), \tag{11}$$

where

$$u(t, \mathbf{X}_V, \widetilde{\mathbf{X}}_V) = \mathrm{MLP}^3\left(\mathrm{MLP}^1(\mathbf{X}_V||t) \ || \ \mathrm{MLP}^2(\widetilde{\mathbf{X}}_V||t)\right)$$
$$v(t, \mathbf{X}_V, \widetilde{\mathbf{X}}_V) = \mathrm{MLP}^4\left(\mathrm{MLP}^1(\mathbf{X}_V||t) \ || \ \mathrm{MLP}^2(\widetilde{\mathbf{X}}_V||t)\right).$$

Here, $\varphi_u$ and $\varphi_v$ are two functions that have a range $[0, 1]$ and attain 0 at the origin. The input $\mathbf{X}$ is split into $\mathbf{X}_U$ and $\mathbf{X}_V$ and the second part goes through the GCN encoder, producing $\widetilde{\mathbf{X}}_V$. Note that one cannot apply the GCN encoder on the entire $\mathbf{X}$; otherwise, the $U$ block will have a dependency on itself in the scaling and the shift. The scaling network $u$ and the shift network $v$ are essentially MLPs that share initial layers; they take both $\mathbf{X}_V$ and $\widetilde{\mathbf{X}}_V$ as inputs.

## 4.4 Learning the Graph

GNeuralFlow contains two sets of parameters: the DAG matrix $\mathbf{A}$ and other parameters of $F$ (call them $\boldsymbol{\theta}$), including the flow parameters, the graph encoder parameters, and possibly other parameters (e.g., in latent variable modeling, Appendix F). Let us use $\mathcal{L}(\mathbf{A}, \boldsymbol{\theta})$ to denote the training loss (which could be the quadratic loss in regression models, or likelihood/ELBO loss in latent variable models), making an explicit distinction between $\mathbf{A}$ and $\boldsymbol{\theta}$. Then, the learning problem is:

$$\min_{\mathbf{A}, \boldsymbol{\theta}} \quad \mathcal{L}(\mathbf{A}, \boldsymbol{\theta}) \quad \text{s.t. } \mathbf{A} \text{ corresponds to a DAG.} \tag{12}$$

The DAG constraint is combinatorial, which makes the problem NP-hard [8]. Fortunately, it is known that $\mathbf{A}$ is a DAG matrix iff $\operatorname{tr}(\operatorname{expm}(\mathbf{A} \odot \mathbf{A})) = n$ or $\operatorname{tr}((\mathbf{I} + \alpha \mathbf{A} \odot \mathbf{A})^n) = n$ for any $\alpha \neq 0$ [48, 47]. Hence, (12) becomes an equality-constrained problem over continuous variables, to which the augmented Lagrangian method [2] is an effective solution. We discuss the optimization details in Appendix D.

# 5 Experiments

We conduct a comprehensive set of experiments to demonstrate that the proposed graph-based approach effectively improves the performance of time series tasks. The experiments are done on four synthetically generated interacting systems and four real-life datasets. Details of these datasets and their tasks are given in Appendix G. In all experiments, we split the data into train, validation, and test sets. We train with early stopping by using Adam and report the results on the test set. Standard errors are obtained by performing five repetitive runs. Hyperparmeter details are given in Appendix H. All experiments are conducted on a machine with an Nvidia A100 GPU, 8 CPU cores, and 80GB main memory.

## 5.1 Synthetic Systems

We generate four synthetic systems with the graph size varying from 3 to 30. These systems follow the graph-based equation (4) or solution (5). They are named "Sink," "Triangle," "Sawtooth," and "Square," following those defined in [4]; but we add a graph to make the system SEM-like. For example, Triangle is generated by following $F(t, \mathbf{X}, \mathbf{A}) = (\mathbf{I} - \mathbf{A}^\top)(\mathbf{X} + \int_0^t \operatorname{sign}(\sin(u)) \, du)$; see Appendix G for other systems and details.

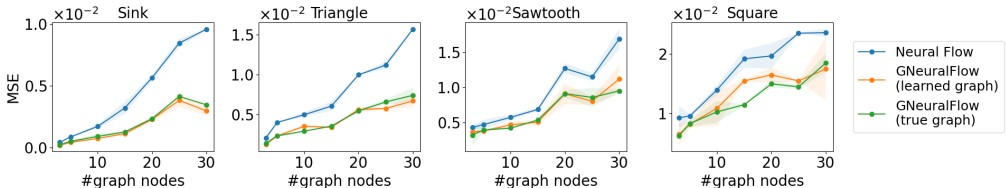

Figure 2: Comparison with neural flow for forecasting on synthetic systems. (ResNet flow).

The forecast results are reported in Figure 2. Two observations follow. First, across datasets and across system sizes, GNeuralFlow lowers the MSE of neural flows. This should not be surprising, since the systems are generated to be interacting and thus standard neural flows that treat the trajectories independently struggle to capture the influence of the graph. Second, GNeuralFlow performs similarly when the graph is either learned or supplied by the ground truth. This observation indicates that the learned graph sufficiently encodes the interacting nature of the trajectories.

We further compare GNeuralFlow with various baselines, including neural ODE, neural flows, graph ODEs (GDE [38], LG-ODE [24], and CF-GODE [26]), graph learning methods (NRI [31] and dNRI [20]), and a non-graph GRU variant for time series (GRU-D [6]). For graph ODEs, the ground-truth graph is used. For neural flows and GNeuralFlow, all three flow designs are experimented with. Table 1 shows that across all datasets, GNeuralFlow significantly outperforms the baselines; moreover, GNeuralFlow also significantly outperforms neural flow for each flow design. These findings indicate that our graph encoder is rather effective and the modeling of conditional dependencies (a DAG structure) is advantageous over that of other graph structures.

Table 1: Comparison with non-graph neural flows/ODE, graph ODE, and other time series methods on synthetic systems (5-node graphs). Best is **boldfaced** and second-best is highlighted in gray.

| | | Sink MSE ($\times 10^{-4}$) | Triangle MSE ($\times 10^{-3}$) | Sawtooth MSE ($\times 10^{-3}$) | Square MSE ($\times 10^{-3}$) |
|---|---|---|---|---|---|
| No Graph | Neural ODE | 10.6 (± 0.03) | 8.32 (± 0.24) | 9.32 (± 0.36) | 16.8 (± 0.39) |
| | Neural flow (ResNet) | 8.41 (± 0.05) | 4.01 (± 0.52) | 4.73 (± 0.06) | 9.61 (± 0.02) |
| | Neural flow (GRU) | 10.9 (± 0.43) | 10.3 (± 0.45) | 16.1 (± 0.41) | 17.2 (± 0.51) |
| | Neural flow (Coupling) | 9.31 (± 0.23) | 12.2 (± 0.41) | 14.2 (± 0.24) | 13.0 (± 0.63) |
| | GRU-D | 12.3 (± 0.23) | 11.3 (± 0.32) | 17.6 (± 0.53) | 18.7 (± 0.31) |
| Graph ODE | GDE | 10.4 (± 0.20) | 3.99 (± 0.05) | 7.65 (± 0.03) | 15.89 (± 0.81) |
| | LG-ODE | 8.57 (± 0.06) | 3.58 (± 0.21) | 7.07 (± 0.02) | 13.99 (± 0.73) |
| | CF-GODE | 8.60 (± 0.14) | 7.19 (± 0.02) | 8.19 (± 0.03) | 13.53 (± 0.11) |
| Graph Learn | NRI | 5.25 (± 0.02) | 3.96 (± 0.16) | 4.99 (± 0.12) | 9.39 (± 0.45) |
| | dNRI | 5.40 (± 0.04) | 3.39 (± 0.09) | 4.97 (± 0.21) | 9.78 (± 0.21) |
| Our Method | GNeuralFlow (ResNet) | **3.95** (± 0.32) | **2.32** (± 0.11) | **3.84** (± 0.06) | **8.24** (± 0.64) |
| | GNeuralFlow (GRU) | 6.83 (± 0.23) | 5.41 (± 0.23) | 5.11 (± 0.13) | 9.14 (± 0.61) |
| | GNeuralFlow (Coupling) | 4.45 (± 0.51) | 3.21 (± 0.34) | 4.25 (± 0.09) | 8.33 (± 0.23) |

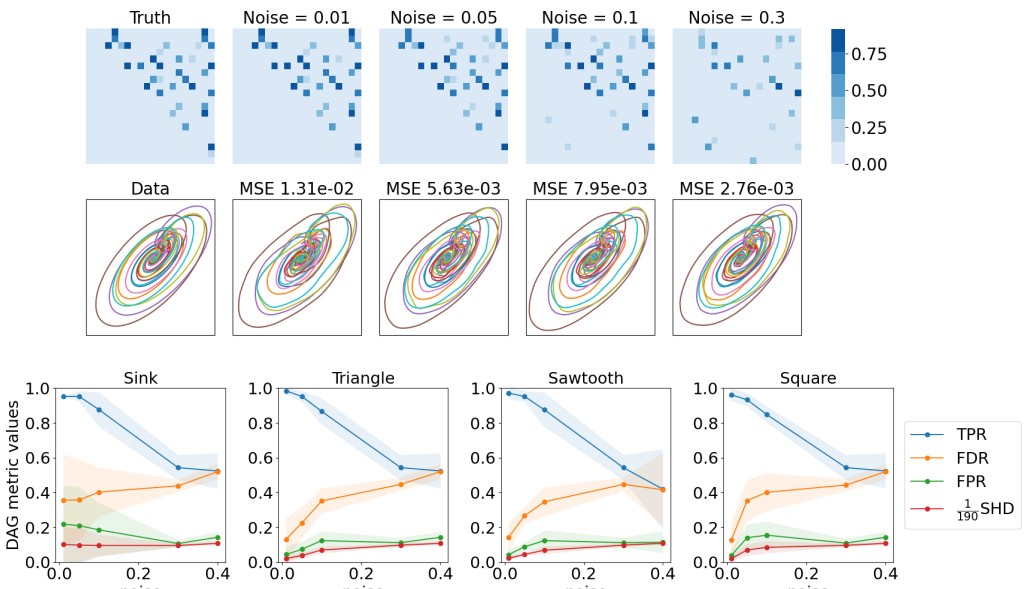

Figure 3: Graph learning quality and forecast quality. Top two rows: Sink (20 nodes); bottom row: all four datasets (20 nodes).

With the above encouraging results, we investigate the quality of graph learning. While the learning approach discussed in Section 4.4 works for a generally initialized **A** (as demonstrated by the previous plots), we perform a more in-depth investigation by initializing **A** through perturbing each entry of the ground truth with a zero-mean Gaussian. We evaluate graph quality by using the metrics proposed by [48]: TPR (true positive rate), FDR (false positive rate), FPR (false prediction rate), and SHD (structural Hamming distance).

Figure 3 shows that as the standard deviation of the Gaussian increases, the learned DAG is more and more different from the ground truth, with the TPR decreasing and the FDR, FPR, and SHD increasing. However, the forecast MSE remains relatively flat (in fact, the MSE from the ground truth is slightly higher). Note that with the nonzeros of the ground truth lying between 0 and 1, a Gaussian with standard deviation 0.3 as the initial guess barely carries the signal of the original graph. We have not been able to establish identifiability conditions for **A** from the ODE (4) as a data generation model; and we suspect that the conditions, if at all exist, may be unrealistically restrictive, given the

empirical findings that a better downstream performance can be achieved by a DAG significantly different from the ground truth. However, even though the ground truth is not recovered, a better downstream performance showcases the robust advantage of a graph-based model that intends to capture the complex interplay inside a system.

We also compare the time costs of neural ODE, neural flows, and GNeuralFlow. Table 2 indicates that GNeuralFlow is more expensive than the corresponding neural flow, because of the additional modeling of the graph. However, GNeuralFlow is more economic than neural ODE, as expected, because it does not run a numerical solver.

Table 2: Time comparison (in seconds) with neural ODE and neural flows on synthetic systems.

|  | Sink | Triangle | Sawtooth | Square |
|---|---|---|---|---|
| Neural ODE | 1.529 | 1.527 | 1.742 | 2.206 |
| Neural flow (ResNet) | 1.022 | 1.013 | 1.021 | 1.020 |
| Neural flow (GRU) | 0.251 | 0.249 | 0.247 | 0.247 |
| Neural flow (Coupling) | 0.136 | 0.137 | 0.136 | 0.133 |
| GNeuralFlow (ResNet) | 1.521 | 1.521 | 1.534 | 1.533 |
| GNeuralFlow (GRU) | 0.275 | 0.283 | 0.286 | 0.284 |
| GNeuralFlow (Coupling) | 1.215 | 1.214 | 1.212 | 1.213 |

## 5.2 Latent Variable Modeling: Smoothing

Just like neural ODE and neural flows, GNeuralFlow can be used for latent variable modeling (see details in Appendix F). To illustrate the effectiveness of GNeuralFlow for this application, we first perform experiments with the smoothing approach in this subsection, by using real-life datasets Activity, Physionet, and MuJoCo [4]. For Activity, we treat each sensor as a graph node; while for the other two, we treat each feature as a node. The tasks are to reconstruct the time series, to classify the activity at each time step (Activity), and to predict the mortality of patients based on the entire time series (Physionet). We again compare our methods with baselines including neural ODE, neural flows, and graph ODEs. For graph ODEs, we construct a dynamic graph at each time step by utilizing the covariance matrix of the time series data within each batch.

Table 3: Comparison with non-graph neural flows/ODE and graph ODE methods for the smoothing approach. Left two: classification task; right three: reconstruction.

|  |  | Activity Accuracy | Physionet AUC | Activity MSE ($\times 10^{-2}$) | Physionet MSE ($\times 10^{-3}$) | MujoCo MSE ($\times 10^{-3}$) |
|---|---|---|---|---|---|---|
| No Graph | ODE-RNN | 0.785 ($\pm$ 0.003) | 0.781 ($\pm$ 0.004) | 6.050 ($\pm$ 0.10) | 4.52 ($\pm$ 0.03) | **2.540** ($\pm$ 0.12) |
| | Neural flow (ResNet) | 0.760 ($\pm$ 0.004) | 0.784 ($\pm$ 0.010) | 6.279 ($\pm$ 0.09) | 4.90 ($\pm$ 0.12) | 8.403 ($\pm$ 0.14) |
| | Neural flow (GRU) | 0.783 ($\pm$ 0.008) | 0.788 ($\pm$ 0.008) | 5.837 ($\pm$ 0.07) | 5.04 ($\pm$ 0.13) | 4.249 ($\pm$ 0.07) |
| | Neural flow (Coupling) | 0.752 ($\pm$ 0.012) | 0.788 ($\pm$ 0.004) | 6.579 ($\pm$ 0.04) | 4.86 ($\pm$ 0.07) | 4.217 ($\pm$ 0.14) |
| Graph ODE | GDE | 0.721 ($\pm$ 0.014) | 0.757 ($\pm$ 0.010) | 6.491 ($\pm$ 0.011) | 4.83 ($\pm$ 0.38) | 5.220 ($\pm$ 0.42) |
| | LG-ODE | 0.743 ($\pm$ 0.023) | 0.748 ($\pm$ 0.018) | 5.738 ($\pm$ 0.089) | 4.87 ($\pm$ 0.27) | 6.699 ($\pm$ 0.83) |
| | CG-ODE | 0.768 ($\pm$ 0.048) | 0.783 ($\pm$ 0.082) | 6.241 ($\pm$ 0.012) | 4.73 ($\pm$ 0.07) | 4.312 ($\pm$ 0.17) |
| Our Method | GNeuralFlow (ResNet) | 0.786 ($\pm$ 0.009) | 0.800 ($\pm$ 0.009) | 5.947 ($\pm$ 0.03) | 4.31 ($\pm$ 0.06) | 2.916 ($\pm$ 0.21) |
| | GNeuralFlow (GRU) | 0.804 ($\pm$ 0.003) | **0.812** ($\pm$ 0.001) | **5.169** ($\pm$ 0.05) | **4.23** ($\pm$ 0.15) | 4.112 ($\pm$ 0.13) |
| | GNeuralFlow (Coupling) | **0.808** ($\pm$ 0.005) | 0.808 ($\pm$ 0.008) | 5.431 ($\pm$ 0.10) | 4.59 ($\pm$ 0.23) | 3.849 ($\pm$ 0.07) |

From Table 3, we see that GNeuralFlow generally performs the best compared with the various baselines. Moreover, by using the same flow design, GNeuralFlow is always better than neural flows. These findings are consistent with those in the synthetic data case, demonstrating a good utility of our method for real-life applications.

## 5.3 Latent Variable Modeling: Filtering

We also perform experiments with the filtering approach, on the MIMIC-IV dataset [4]. We treat each feature as a node to set up a graph of 97 longitudinal features, including lab tests, outputs, and prescriptions in clinical events. This graph is the largest among all experiments in this paper.

Table 4 reports the forecast error on the next three time points and the estimated likelihood of the time series. We see that GNeuralFlow performs the best with the GRU flow design, while some graph ODE approaches come second. Moreover, GNeuralFlow consistently outperforms neural flows, across flow architecture designs and evaluation metrics. The improvement over corresponding neural flow versions is at least the sum of the standard errors of the two compared methods, which are demonstratively significant.

Table 4: Comparison with non-graph neural flows/ODE and graph ODE methods for the filtering approach. For both metrics, the lower the better.

|  |  | MSE | NLL |
|---|---|---|---|
| No Graph | GRU-ODE-Bayes | 0.379 ($\pm$ 0.005) | 0.748 ($\pm$ 0.045) |
|  | Neural flow (ResNet) | 0.379 ($\pm$ 0.005) | 0.774 ($\pm$ 0.059) |
|  | Neural flow (GRU) | 0.364 ($\pm$ 0.008) | 0.734 ($\pm$ 0.054) |
|  | Neural flow (Coupling) | 0.366 ($\pm$ 0.002) | 0.675 ($\pm$ 0.003) |
| Graph ODE | GDE | 0.342 ($\pm$ 0.001) | 0.657 ($\pm$ 0.007) |
|  | LG-ODE | 0.349 ($\pm$ 0.002) | 0.649 ($\pm$ 0.005) |
|  | CG-ODE | 0.372 ($\pm$ 0.011) | 0.825 ($\pm$ 0.018) |
| Our Method | GNeuralFlow (ResNet) | 0.356 ($\pm$ 0.0007) | 0.663 ($\pm$ 0.008) |
|  | GNeuralFlow (GRU) | **0.335** ($\pm$ 0.003) | **0.606** ($\pm$ 0.001) |
|  | GNeuralFlow (Coupling) | 0.350 ($\pm$ 0.004) | 0.662 ($\pm$ 0.008) |

## 6 Conclusions and Discussions

In this work, we address the challenge of learning the systemic interactions of time series, by proposing a graph-based model GNeuralFlow and learning the graph structure in tandem with the system dynamics. GNeuralFlow is a continuous-time model, which can be used for irregularly sampled time series. Moreover, the systemic interactions are modeled by a conditional dependence structure. We apply GNeuralFlow to latent variable modeling and demonstrate that incorporating the DAG structure improves time series classification and forecasting noticeably.

Several time-series approaches do not learn a static graph but a dynamic one [20]. Such a graph is interpreted as a latent structure, which varies depending on past data. In contrast, our approach learns an explicit structure that governs the dynamics over time. Nevertheless, our mathematical framework can be straightforwardly adapted to learning a time-varying latent graph, if desired. To achieve so, we reuse the loss calculation in (12), remove the constraint, and parameterize $\mathbf{A}$ as a function of $\mathbf{X}(t)$. This way, we sacrifice the DAG interpretation of the interactions but gain a time-dependent graph.

A limitation of the proposed model is that the number of parameters on the $\mathbf{A}$ part grows quadratically with the number of time series (nodes). Hence, this part of the computational cost can be cubic, because the evaluation of the DAG constraint and the gradient involves the computation of the matrix exponential. Such a scalability challenge is a common problem for DAG structure learning. While past research showcased the feasibility of learning a graph with a few hundred nodes [47], going beyond is generally believed to require either a new computational technique or a new modeling approach. One potential direction is to introduce structures into $\mathbf{A}$ (such as low-rankness [15]), which admit faster matrix evaluation.

## Acknowledgments and Disclosure of Funding

GM acknowledges support from the Engineering and Physical Sciences Research Council (EPSRC) and the BBC under iCASE. AF is partially funded by the CRUK National Biomarker Centre, by the Manchester Experimental Cancer Medicine Centre and the NIHR Manchester Biomedical Research Centre. JC is supported by the MIT-IBM Watson AI Lab.

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

## A  Supporting Code

Code is available at `https://github.com/gmerca/GNeuralFlow`.

## B  Proofs

*Proof of Theorem 1.*  When $\mathbf{A}$ is a DAG adjacency matrix, its symmetrization $\mathbf{B}$ is the adjacency matrix of the corresponding undirected graph. We call the DAG an orientation of the undirected graph. The Gershgorin circle theorem asserts that the spectral radius of $\mathbf{B}$, $\rho(\mathbf{B})$, is bounded by $\gamma$. Meanwhile, [23] show that the spectral norm $\|\mathbf{A}\|_2 \leq \rho(\mathbf{B})$. Then, $\|\widehat{\mathbf{A}}\|_2 \leq 1 + \gamma/\gamma = 2$.  □

*Proof of Theorem 2.*  We follow the proof of [4, Theorem 1] (see A.3 of the paper), which concludes that $|h(x) - h(y)| \leq \alpha(\frac{5}{4}\beta + \frac{3}{2})|x - y|$. Since GCN is contractive, such an inequality applies to both $h = h^1$ and $h = h^2$. Then, applying Eqn (14) of the paper,

$$|h^1(x)h^2(x) - h^1(y)h^2(y)| < \big[ \underbrace{|h^1(x)|}_{<1} \cdot \underbrace{\mathrm{Lip}(h^2)}_{<\alpha(\frac{5}{4}\beta+\frac{3}{2})} + \underbrace{|h^2(x)|}_{<1} \cdot \underbrace{\mathrm{Lip}(h^1)}_{<\alpha(\frac{5}{4}\beta+\frac{3}{2})} \big]|x-y| < 2\alpha(\tfrac{5}{4}\beta+\tfrac{3}{2})|x-y|.$$

Therefore, when $\alpha(5\beta + 6) \leq 2$, the product $h^1 h^2$ is contractive and therefore $F$ is invertible.  □

## C  Details of Example in Section 3

The matrix is

$$\mathbf{B} = \begin{bmatrix} -4 & 5 \\ -3 & 1 \end{bmatrix},$$

and the initial conditions are

$$\mathbf{x}_0^1 = \begin{bmatrix} 0.6 \\ 0.5 \end{bmatrix}, \quad \mathbf{x}_0^2 = \begin{bmatrix} 0.7 \\ 0.1 \end{bmatrix}, \quad \mathbf{x}_0^3 = \begin{bmatrix} 0.2 \\ 0.3 \end{bmatrix}.$$

## D  Training Method

In this section, we briefly describe the augmented Lagrangian method for solving the equality-constrained problem

$$\min_{\mathbf{A},\boldsymbol{\theta}} \quad \mathcal{L}(\mathbf{A}, \boldsymbol{\theta})$$
$$\text{s.t.} \quad h(\mathbf{A}) = 0,$$

where the constraint can either be $h(\mathbf{A}) = \mathrm{tr}(\mathrm{expm}(\mathbf{A} \odot \mathbf{A})) - n$ or $h(\mathbf{A}) = \mathrm{tr}((\mathbf{I} + \alpha\mathbf{A} \odot \mathbf{A})^n) - n$.

Define the augmented Lagrangian

$$\mathcal{L}_c = \mathcal{L}(\mathbf{A}, \boldsymbol{\theta}) + \lambda h(\mathbf{A}) + \frac{c}{2}|h(\mathbf{A})|^2, \tag{13}$$

where $\lambda$ and $c$ denote the Lagrange multiplier and the penalty parameter, respectively. The general idea of the method is to gradually increase the penalty parameter to ensure that the constraint is eventually satisfied. Over iterations, $\lambda$ as a dual variable will converge to the Lagrange multiplier of the original problem. The upate rule at the $k$th iteration reads

$$\mathbf{A}^k, \boldsymbol{\theta}^k = \operatorname*{argmin}_{\mathbf{A},\boldsymbol{\theta}} \mathcal{L}_{c^k}$$
$$\lambda^{k+1} = \lambda^k + c^k h(\mathbf{A}^k)$$
$$c^{k+1} = \begin{cases} \eta c^k & \text{if } |h(\mathbf{A}^k)| > \gamma|h(\mathbf{A}^{k-1})| \\ c^k & \text{else,} \end{cases}$$

where $\eta \in (1, +\infty)$ and $\gamma \in (0, 1)$ are hyperparameters to be tuned.

The subproblem of optimizing $\mathbf{A}$ and $\boldsymbol{\theta}$ can be solved by using the Adam optimizer. It requires the gradient of $\mathcal{L}_c$ and hence of $h$. For $h(\mathbf{A}) = \text{tr}(\text{expm}(\mathbf{A} \odot \mathbf{A})) - n$, it can be derived that $\nabla h(\mathbf{A}) = \text{expm}(\mathbf{A} \odot \mathbf{A})^\top \odot 2\mathbf{A}$, which can be obtained virtually for free after $h$ has been evaluated. For $h(\mathbf{A}) = \text{tr}((\mathbf{I} + \alpha \mathbf{A} \odot \mathbf{A})^n) - n$, one may use automatic differentiation to obtain the gradient.

Algorithm 1 summarizes the training procedure. Note that an effective initialization of $\mathbf{A}$ would use an empty diagonal. Moreover, in every update of $\mathbf{A}$, one may keep its diagonal zero throughout.

---
**Algorithm 1** Training algorithm of GNeuralFlow
---
1: Initialize $c \leftarrow 1$ and $\lambda \leftarrow 0$
2: **for** $k = 0, 1, 2, \ldots$ **do**
3:     Compute $\mathbf{A}^k$ and $\boldsymbol{\theta}^k$ as a minimizer of (13) by using the Adam optimizer
4:     Update Lagrange multiplier $\lambda \leftarrow \lambda + ch(\mathbf{A}^k)$
5:     **if** $k > 0$ and $|h(\mathbf{A}^k)| > \gamma|h(\mathbf{A}^{k-1})|$ **then**
6:         $c \leftarrow \eta c$
7:     **end if**
8:     **if** $|h(\mathbf{A}^k)| <$ threshold **then**
9:         break
10:    **end if**
11: **end for**
---

## E  Handling Missing Data

At a particular time $t$, some rows of the observed data $\mathbf{X}(t)$ may be empty (i.e., measurements of some time series at time $t$ are missing). In this case, one evaluates the graph neural flow $F$ on a subgraph of present measurements. Rather than extracting this subgraph and the corresponding rows of $\mathbf{X}$, the GCN encoder (6) offers a convenient approach for evaluation: masking. This approach is particularly favorable in batching, because tensor dimensions do not change. In particular, we mask out (i.e., setting zero) the part of $\mathbf{A}$ corresponding to missing data. Then, for the output $\widetilde{\mathbf{X}}$, the part corresponding to present data is correctly calculated, while the part of missing data becomes zero and this condition is invariant across layers. Note that the GCN layers must not have bias terms to maintain this invariance.

## F  GNeuralFlow for Latent Variable Modeling

In addition to straightforwardly modeling the data space, neural flows find successful use in the latent space. Here, we discuss two popular approaches in latent variable modeling and how GNeuralFlow can be incorporated.

Of particular consideration is the role of the graph. One may straightforwardly extend [4] by using the flow in the latent/hidden space; however, this method models the interactions among the latent/hidden dimensions, which are less interpretable than those among the time series. Hence, we propose to use the flow on an augmented space, part of which carries the graph information, as a new design complementary to those proposed in Section 4.3.

On a high level, smoothing [39] and filtering [5] approaches use a neural ODE or a neural flow to continuously evolve the hidden state from time $t_{j-1}$ (denoted as $\mathbf{H}(t_{j-1})$) to time $t_j$ (denoted as $\mathbf{H}'(t_j)$); and then use an RNN to introduce a jump (denoted as $\mathbf{H}(t_j)$) on observing input data $\mathbf{X}(t_j)$. To model and learn the interaction graph in the data space, we use the graph encoder (6) to produce a transformed data $\widetilde{\mathbf{X}}(t_j)$ and use a second RNN to introduce the paired jump $\widetilde{\mathbf{H}}(t_j)$ given $\widetilde{\mathbf{X}}(t_j)$. Then, the pair of hidden states, $\mathbf{H}(t_j)$ and $\widetilde{\mathbf{H}}(t_j)$, are concatenated and a standard neural flow evolve the concatenated state to the next time point, the result of which is then projected to the proper hidden dimension. The smoothing and filtering approaches differ in fine details, including different uses of the RNNs and hidden states. As a result, the loss function $\mathcal{L}$ in (12) is also different. Details are presented in the following.

## F.1 Smoothing Approach

Given observation data $\mathbf{X}(t_0), \ldots, \mathbf{X}(t_N)$, this approach produces a latent quantity $\mathbf{Z}_0$ by a combined use of LSTM and neural flow, and then traces out a smooth curve $\mathbf{Z}(t)$ using another flow, taking $\mathbf{Z}(t_0) = \mathbf{Z}_0$ as the initial condition. Then, the observation data $\mathbf{X}(t_j)$ is recovered from $\mathbf{Z}(t_j)$.

A VAE is used to set up the training loss. The decoder $p(\mathbf{X}(t_0), \ldots, \mathbf{X}(t_N)|\mathbf{Z}_0)$ is factorized as

$$p(\mathbf{X}(t_0), \ldots, \mathbf{X}(t_N)|\mathbf{Z}_0) = \prod_{j=0}^{N} p(\mathbf{X}(t_j)|\mathbf{Z}(t_j)),$$

where each $\mathbf{Z}(t_j)$ is computed by running a standard neural flow: $\mathbf{Z}(t_j) = F(t_j, \mathbf{Z}_0)$. The encoder, on the other hand, produces a latent Gaussian $\mathbf{Z}_0$ with mean $\boldsymbol{\mu}$ and diagonal covariance $\mathrm{diag}(\boldsymbol{\sigma})$; that is,

$$q(\mathbf{Z}_0|\mathbf{X}(t_0), \ldots, \mathbf{X}(t_N)) = \mathcal{N}(\mathbf{Z}_0 \,|\, \boldsymbol{\mu}, \mathrm{diag}(\boldsymbol{\sigma})), \quad [\boldsymbol{\mu}, \log \boldsymbol{\sigma}] = g(\mathbf{H}(t_N)),$$

where $\mathbf{H}(t_N)$ is the hidden state to be elaborated soon and $g$ is a neural network projection.[3] The ELBO loss for the VAE is

$$\mathcal{L}_{\mathrm{ELBO}} = D_{\mathrm{KL}}\Big(q(\mathbf{Z}_0|\mathbf{X}(t_0), \ldots, \mathbf{X}(t_N)) \,\|\, p(\mathbf{Z}_0)\Big)$$
$$- \mathrm{E}_{\mathbf{Z}_0 \sim q(\mathbf{Z}_0|\mathbf{X}(t_0), \ldots, \mathbf{X}(t_N))} \Big[\log p(\mathbf{X}(t_0), \ldots, \mathbf{X}(t_N)|\mathbf{Z}_0)\Big].$$

Our GNeuralFlow uses a pair of LSTMs together with another neural flow to evolve the hidden state. Specifically, we maintain a pair of states $\mathbf{H}(t)$ and $\widetilde{\mathbf{H}}(t)$, the latter of which includes the graph information. At time $t_{j-1}$, we concatenate $\mathbf{H}(t_{j-1})$ and $\widetilde{\mathbf{H}}(t_{j-1})$, run the neural flow $F$ to evolve the concatenated state to time $t_j$, and apply a projection $g_{\mathrm{proj}}$ so that the net result $\mathbf{H}'(t_j)$ remains in the same dimension as $\mathbf{H}(t_{j-1})$:

$$\mathbf{H}'(t_j) = g_{\mathrm{proj}}\big( F(t_j, \mathbf{H}(t_{j-1})\|\widetilde{\mathbf{H}}(t_{j-1})) \big).$$

Then, we run a pair of LSTMs to obtain the paired states at time $t_j$:

$$\mathbf{H}(t_j) = \mathrm{LSTM}^1\big(\mathbf{H}'(t_j), \ \mathbf{X}(t_j)\big), \quad \widetilde{\mathbf{H}}(t_j) = \mathrm{LSTM}^2\big(\mathbf{H}'(t_j), \ \widetilde{\mathbf{X}}(t_j)\big),$$

where the second LSTM is applied to the transformed observation data $\widetilde{\mathbf{X}}(t_j)$ produced by the GCN encoder (6). By doing so, the graph models the interaction inside the data $\mathbf{X}(t)$ rather than the hidden states $\mathbf{H}(t)$.

## F.2 Filtering Approach

As opposed to the preceding approach, the filtering approach uses only a decoder. Each time, it first evolves the hidden state to $\mathbf{H}'(t_j)$ and then runs a GRU to update the hidden state to $\mathbf{H}(t_j)$. This approach maintains two Gaussians, the first one models the observation $\mathbf{X}(t_j)$:

$$\mathcal{N}(\mathbf{X}(t_j) \,|\, \boldsymbol{\mu}_{\mathrm{obs}}^j, \mathrm{diag}(\boldsymbol{\sigma}_{\mathrm{obs}}^j)), \quad [\boldsymbol{\mu}_{\mathrm{obs}}^j, \log \boldsymbol{\sigma}_{\mathrm{obs}}^j] = g_{\mathrm{obs}}(\mathbf{H}'(t_j)),$$

while the second one models the jump caused by the GRU:

$$\mathcal{N}(\boldsymbol{\mu}_{\mathrm{post}}^j, \mathrm{diag}(\boldsymbol{\sigma}_{\mathrm{post}}^j)), \quad [\boldsymbol{\mu}_{\mathrm{post}}^j, \log \boldsymbol{\sigma}_{\mathrm{post}}^j] = g_{\mathrm{post}}(\mathbf{H}(t_j)).$$

The training loss aims at maximizing the observation data likelihood while minimizing the KL divergence of the two Gaussians:

$$\mathcal{L} = -\sum_{j=1}^{N} \log \mathcal{N}(\mathbf{X}(t_j) \,|\, \boldsymbol{\mu}_{\mathrm{obs}}^j, \mathrm{diag}(\boldsymbol{\sigma}_{\mathrm{obs}}^j)) + \lambda D_{\mathrm{KL}}\Big(\mathcal{N}(\boldsymbol{\mu}_{\mathrm{obs}}^j, \mathrm{diag}(\boldsymbol{\sigma}_{\mathrm{obs}}^j)) \,\|\, \mathcal{N}(\boldsymbol{\mu}_{\mathrm{post}}^j, \mathrm{diag}(\boldsymbol{\sigma}_{\mathrm{post}}^j))\Big).$$

Our GNeuralFlow uses a pair of GRUs together with a standard neural flow to evolve the hidden state. Specifically, we maintain a pair of states $\mathbf{H}(t)$ and $\widetilde{\mathbf{H}}(t)$, the latter of which includes the graph

---

[3]When the encoder is run backward in time, one uses $\mathbf{H}(t_0)$ instead of $\mathbf{H}(t_0)$.

information. At time $t_{j-1}$, we concatenate $\mathbf{H}(t_{j-1})$ and $\widetilde{\mathbf{H}}(t_{j-1})$, run the neural flow $F$ to evolve the concatenated state to time $t_j$, and apply a projection $g_{\text{proj}}$ so that the net result $\mathbf{H}'(t_j)$ remains in the same dimension as $\mathbf{H}(t_{j-1})$:

$$\mathbf{H}'(t_j) = g_{\text{proj}}\big( F(t_j, \mathbf{H}(t_{j-1})||\widetilde{\mathbf{H}}(t_{j-1})) \big).$$

Then, we run a pair of GRUs to obtain the paired states at time $t_j$:

$$\mathbf{H}(t_j) = \text{GRU}^1\Big(\mathbf{H}'(t_j),\ g_{\text{prep}}(\mathbf{X}(t_j), \mathbf{H}'(t_j))\Big), \quad \widetilde{\mathbf{H}}(t_j) = \text{GRU}^2\Big(\mathbf{H}'(t_j),\ g_{\text{prep}}(\widetilde{\mathbf{X}}(t_j), \mathbf{H}'(t_j))\Big),$$

where the second GRU is applied to the transformed observation data $\widetilde{\mathbf{X}}(t_j)$ produced by the GCN encoder (6). By doing so, the graph models the interaction inside the data $\mathbf{X}(t)$ rather than the hidden states $\mathbf{H}(t)$.

# G    Datasets and Tasks

Table 5: Experiment settings and datasets.

| Dataset | Method | Tasks & Metrics | #Nodes ($n$) | #Times ($N$) | #Samples | Split |
|---|---|---|---|---|---|---|
| Synthetic | regression | forecast MSE | 5–30 | 500 | 1000 | 60:20:20 |
| Synthetic | regression | graph metrics | 15 | 500 | 1000 | 60:20:20 |
| Activity | smoothing | reconstruction MSE classification accuracy | 4 | 50 | 6554 | 75:5:20 |
| Physionet | smoothing | reconstruction MSE classification AUC | 41 | 52 | 8000 | 60:20:20 |
| MuJoCo | smoothing | reconstruction MSE | 14 | 100 | 10000 | 60:20:20 |
| MIMIC-IV | filtering | forecast MSE log-likelihood | 97 | 19 | 17874 | 70:15:15 |

The datasets used in this paper include four synthetic ODE systems and four real-life datasets. Table 5 summarizes the basic information of these datasets, tasks, evaluation metrics, and learning methods.

## G.1    Synthetic Datasets

We define four interacting systems based on either the ODE $\dot{\mathbf{X}} = f(t, \mathbf{X}, \mathbf{A})$ or the solution $\mathbf{X}(t) = F(t, \mathbf{X}_0, \mathbf{A})$:

- Sink (2D): $f(t, \mathbf{X}, \mathbf{A}) = (\mathbf{I} - \mathbf{A}^\top)\mathbf{X}\mathbf{B}^\top$ where $\mathbf{B} = \begin{bmatrix} -4 & 10 \\ -3 & 2 \end{bmatrix}$
- Triangle (1D): $F(t, \mathbf{X}, \mathbf{A}) = (\mathbf{I} - \mathbf{A}^\top)(\mathbf{X} + \int_0^t \text{sign}(\sin(u))\, du)$
- Sawtooth (1D): $F(t, \mathbf{X}, \mathbf{A}) = (\mathbf{I} - \mathbf{A}^\top)(\mathbf{X} + t - \lfloor t \rfloor)$
- Square (1D): $F(t, \mathbf{X}, \mathbf{A}) = (\mathbf{I} - \mathbf{A}^\top)(\mathbf{X} + \text{sign}(\sin(t)))$

For Sink, $\mathbf{X} \in \mathbb{R}^{n \times 2}$; while for the other three systems, $\mathbf{X} \in \mathbb{R}^{n \times 1}$, where $n$ is the number of trajectories (graph nodes) in the system. The initial condition $\mathbf{X}_0$ is uniformly sampled from $[0, 1]^{n \times 2}$ for Sink, and from $[-2, 2]^{n \times 1}$ for the other three systems. The time interval is $[0, 10]$ and the time points are uniformly random.

The DAG adjacency matrix is generated by using the following procedure:

1. Generate a sparse $n \times n$ matrix $\mathbf{A}$ with a pre-defined density, where the nonzero locations are random and the nonzero values are uniformly random.
2. Keep only the strict upper triangular part of $\mathbf{A}$ (i.e., diagonal is zero).
3. Perform symmetric row/column permutation on $\mathbf{A}$.

The task is to predict the trajectories $\mathbf{X}(t)$ given $\mathbf{X}_0$.

## G.2 Real-Life Datasets

We use four real-life datasets preprocessed by [4].

Activity [39] contains time series recorded by four sensors, on individuals performing various activities: walking, sitting, lying, etc. The task is to classify the activities at each time point. Additionally, since the smoothing approach for latent variable modeling reconstructs the time series, we also evaluate different models on the reconstruction quality. We treat each sensor as one graph node.

Physionet [42] contains time series of patients' measurements (37 variables in total) from the first 48 hours after being admitted to ICU. The task is to predict the mortality of the patients. Additionally, since the smoothing approach for latent variable modeling reconstructs the time series, we also evaluate different models on the reconstruction quality. We treat each variable as one graph node.

MuJoCo [44] contains physics simulations by randomly sampling initial positions and velocities and letting the dynamics evolve deterministically in time. Each sequence includes 14 features. We treat each feature as one graph node. We evaluate different models on the reconstruction quality.

MIMIC-IV [18, 28] contains time series of ICU patients' measurements, including their vital signs, laboratory test results, medication, and any output data during their ICU stay (97 variables in total). The task is to predict the next three measurements in the 12 hour interval after the observation window of 36 hours. We treat each variable as one graph node.

# H  Hyperparameter Details

Table 6: Graph learning hyperparameters.

| Synthetic systems (all architectures) | | |
| --- | --- | --- |
| # points | $\eta$ | $\gamma$ |
| 3 | 3 | 0.3 |
| 5 | 5 | 0.25 |
| 15 | 7 | 0.21 |
| 20 | 7 | 0.19 |
| 25 | 7 | 0.19 |
| 30 | 7 | 0.16 |

| Real-life datasets | ResNet | | GRU | | Coupling | |
| --- | --- | --- | --- | --- | --- | --- |
| | $\eta$ | $\gamma$ | $\eta$ | $\gamma$ | $\eta$ | $\gamma$ |
| Activity | 7 | 0.21 | 15 | 0.21 | 7 | 0.21 |
| Physionet | 10 | 0.5 | 15 | 0.5 | 10 | 0.5 |
| MuJoCo | 15 | 0.5 | 10 | 0.5 | 15 | 0.5 |
| MIMIC-IV | 10 | 0.15 | 10 | 0.15 | 10 | 0.15 |

We reuse the architecture parameters and training hyperparameters in [4] and only tune the graph learning hyperparameters (see Algorithm 1 in Section D). Table 6 lists the tuned values.

