# OpenReview forum: "Graph Neural Flows for Unveiling Systemic Interactions Among Irregularly Sampled Time Series"
_NeurIPS.cc/2024/Conference — NeurIPS 2024 poster_

### Official Review · Reviewer_2pqA · 2024-06-30

**Soundness:** 4
**Presentation:** 3
**Contribution:** 3
**Rating:** 8
**Confidence:** 4

**Summary:**

Authors propose a novel idea to learn the interactions between time series which is crucial to make reliable forecasts in interacting systems. Paper proposes a new algorithm GNeuralFlow; a new graph-based continuous time model to learn systematic interaction. A GNN is used to learn the interactions between time series, where each time series is associated with a directed acyclic graph. GNeuralFlow learns the graph structure for irregularly sampled time series.

The experiments performed on synthetic and real data shows they GNeuralFlow outperform the other models based regression and classification tasks. The experiments to model latent variable modelling shows GNeuralFlow's robustness in learning and utilizing the underlying graph structure for improved predictive performance.

**Strengths:**

Originality:  Authors identify an important problem in non-uniform time samples time series to model the interactions between them. The paper introduces the original idea (as per my knowledge) of using Graph Neural Flow (GNeuralFlow) which leverages the power of graph based modelling and continuous time technique to capture the relationship between time series.   The method has wide application in climate, finance and more where a dynamic system is crucial.

Quality:   The main development of paper GNeuralFlow is built on a strong mathematical foundation and authors provide empirical and experimental results to support their claims.

Clarity: The paper is well structured and describes the problem clearly and then proceeds to describe GNeuralFlow as a solution to it and advance the field. The text provides a clear explanation of the idea with a complete mathematical framework to support the same.

Significance: The idea of capturing relationship between time series using GNeural is significant improvement in the methodology. Learning the graph as a solution to ODE gives the flexibility to use it for unequal samples of time series. This has several applications in time series domain including streaming data.

The idea is to use traditional GCN to learn the interaction between time series for continuous time GNeuralFlow can handle the time series data observed at non-uniform time stamps without the need for interpolation or computing expensive pre-processing.

**Weaknesses:**

Although the paper address a big problem faced in steaming data application with unequal time samples using novel approach the idea comes with additional complexity. The complex nature of the solution is computationally expensive and is not scalable after a certain size.  The fact has been identified by authors as well and documented in the limitations sections.

**Questions:**

- The paper discusses about the increase in the complexity due to added Graphs in the model. Is there an estimate how much is the computation time increase (is it quadratic?) ? Does this time power increase (from quadratic to cubic) as # of interacting time series are increased in the experiment?

**Limitations:**

The paper is well structured but the biggest strength becomes its limitation as well. The addition of graph and learning the parameters allow the measurement of interactions but at the same time it increase the complexity of the model. This would be a the main challenge to use the model in real applications and might need approximate solutions to adapt this version of paper.

---

> ### Author Rebuttal · Authors · 2024-08-07
>
> Thank you for the thoughtful comments, we reply to your questions below.
>
> **RE: computational cost**
>
> The computational cost can be cubic for the number of nodes because the evaluation of the DAG constraint and the gradient involves the computation of the matrix exponential. One possible mitigation is that rather than modeling the graph adjacency matrix as a free variable, we impose structures on the matrix (such as diagonal plus low rank). Then, the matrix exponential is less expensive to compute.

---

### Official Review · Reviewer_zrVL · 2024-07-12

**Soundness:** 3
**Presentation:** 3
**Contribution:** 3
**Rating:** 3
**Confidence:** 3

**Summary:**

This paper proposes a novel graph-based continuous-time model GNeuralFlow for learning systemic interactions and interactions are modeled as a Bayesian network. Experimental results for regression problems show that the proposed GNF achieves state-of-the-art performances on time series classification and forecasting benchmarks.

**Strengths:**

+ The paper is well-written and easy to follow.
+ The introduction give a nice overview and motivation for the problem.
+ The results are good and evaluation is reasonable.

**Weaknesses:**

- The proposed method lacks a theoretical analysis.
- The authors can consider using state-of-the-art GNN as backbone for graph encoder.
- Can the authors apply this method to real-world datasets?
- Computational complexity and corresponding comparisons with baselines are missing.

**Questions:**

Please see comments in Weaknesses.

**Limitations:**

Not applicable.

---

> ### Author Rebuttal · Authors · 2024-08-07
>
> **RE: theoretical analysis**
>
> Our work mainly focuses on modeling. We offered certain analyses to justify some modeling choices, including guaranteeing contractive mappings required by the neural flows (Theorem 1, Theorem 2, and other inline text within section 4.3).
>
> **RE: sota GNN**
>
> Thank you for pointing out this possible direction. We did not explore further as the network we used was already able to outperform the considered baselines.
>
>
> **RE: real-word dataset**
>
> Thanks for this request. We note that we already performed experiments over four real-life datasets, including Activity, Physionet, Mujoco, and Mimic-IV. See our paper in sections 5.2 and 5.3, and Appendix F for dataset description.
>
>
>
> **RE: computational complexity**
>
> Thanks for this point. We have added a table with the wall-clock runtime. As expected, the graph-neural-flow (GNF) models are more expensive than the neural-flow corresponding ones (because of the additional modeling of the graph). However, the GNF are still cheaper than the Neural ODEs (because we do not need a numerical solver).
>
>
> | Model               | Sink  | Triangle | Sawtooth | Square |
> | ------------------- | ----- | -------- | -------- | ------ |
> | Neural-ODE          | 1.529 | 1.527    | 1.742    | 2.206  |
> | NF-resnet           | 1.022 | 1.013    | 1.021    | 1.020  |
> | NF-coupling         | 0.136 | 0.137    | 0.136    | 0.133  |
> | NF-gru              | 0.251 | 0.249    | 0.247    | 0.247  |
> | GNF-resnet (ours)   | 1.521 | 1.521    | 1.534    | 1.533  |
> | GNF-coupling (ours) | 1.215 | 1.214    | 1.212    | 1.213  |
> | GNF-gru (ours)      | 0.275 | 0.283    | 0.286    | 0.284  |
>
>
> **Final comment**
>
> Thank you for your review. Please consider raising the score if we have addressed your points appropriately.

---

> > ### Comment · Reviewer_zrVL · 2024-08-12
> > **Thank you.**
> >
> > I appreciate the author for the detailed response. After carefully reading the rebuttal, I am retaining my score.

---

> > > ### Author Response · Authors · 2024-08-13
> > >
> > > Dear Reviewer zrVL,
> > >
> > > Thank you for your valuable suggestions to enhance our work. In response to your request for further improvement, we have explored an additional GNN model. Our findings in the table below indicate that significant improvement can be achieved by combining the ResNet model with a Message Passing GNN (Veličković, 2022; Bronstein et al., 2021).
> > >
> > >
> > > | model             | Sink            | Triangle        | Sawtooth        | Square          |
> > > | ----------------- | --------------- | --------------- | --------------- | --------------- |
> > > | GNF-resnet (ours) | 3.95(±0.32)     | 2.32(±0.11)     | 3.84(±0.06)     | 8.24(±0.64)     |
> > > | GNF-MPGNN (ours)  | **3.45(±0.13)** | **2.00(±0.03)** | **2.79(±0.05)** | **4.32(±0.12)** |
> > >
> > > We kindly ask you to consider raising the score if this update, or any of our previous revisions, addresses your concerns either fully or partially.
> > >
> > >
> > > **References**
> > >
> > > Veličković 2022. Message passing all the way up. ICLR
> > >
> > > Bronstein et al 2021. Geometric deep learning. Arxiv.

---

### Official Review · Reviewer_3M7x · 2024-07-12

**Soundness:** 3
**Presentation:** 3
**Contribution:** 2
**Rating:** 6
**Confidence:** 2

**Summary:**

The paper focuses on a graph-based model to capture dependencies in irregularly sampled time series data. The framework employs a causal prior—a directed acyclic graph—where nodes are conditionally independent of non-descendants given their parents, specifying component dynamics dependencies. The proposed model, termed a graph neural flow, learns the solution of an unknown ODE directly from irregularly sampled time series, which contrasts with neural ODEs by avoiding repeated calls to a costly numerical solver. Multiple neural flows, one for each time series, are conditioned on the DAG, with their interactions instantiated using a GNN, such as a graph convolutional network (GCN). The GCN enhances the ODE solution parameterization by aggregating neighboring time series information at each time point, effectively modeling a graph-conditioned ODE that captures system interactions.

**Strengths:**

The proposed model is nicely presented, along with precise and meaningful mathematical formulations for capturing interactions in irregularly sampled time series. GNeuralFlow shows significant performance in both classification and regression problems spanning several synthetic and real-world datasets. Hyperparameters and models’ configurations are presented enabling reproducibility and fair comparisons.

**Weaknesses:**

1. *W1:* Except for the standard experiments on synthetic data representing dynamical systems, the authors explore different tasks on various datasets, slightly deviating from some setups followed in existing related works. For instance, some studies include additional experiments on interpolation/extrapolation [1] or use additional datasets (e.g., classification on MIMIC-III). Moreover, incorporating additional standard baselines in addition to the presented ones could strengthen the evaluation (recurrent/attention-based ones, e.g., mTAND, GRU-D, and others).
2. *W2:* The proposed GNeuralFlow is primarily based on the concept of Neural Flows for efficient computation of ODE system solutions. The significant methodological extensions in the paper include the incorporation of graph-based representations and causal formulation. However, these extensions are not qualitatively evaluated by analyzing causal relationships or visualizing interactions within the learned graphs. For instance, authors in [2] present studies/experiments to quantify causal discovery in the studied datasets.
3. *W3:* Although the approach of replacing the ODE solver offers a computational advantage over classical Neural-ODE-based methods, a comprehensive computational cost analysis comparing different methods, including simpler (e.g., sequential, non-ODE-based) yet effective models, is crucial.
4. *W4:* The presentation of the related work is quite brief, causing some confusion, particularly regarding closely related graph-based methods like CF-GODE and LG-ODE. Including comparisons and comments on additional relevant papers could be beneficial [3,4] (please explain if not relevant).

[1] Schirmer, M., Eltayeb, M., Lessmann, S., & Rudolph, M. (2022, June). Modeling irregular time series with continuous recurrent units. In International conference on machine learning (pp. 19388-19405). PMLR.

[2] Löwe, S., Madras, D., Zemel, R., & Welling, M. (2022, June). Amortized causal discovery: Learning to infer causal graphs from time-series data. In Conference on Causal Learning and Reasoning (pp. 509-525). PMLR.

[3] Choi, J., Choi, H., Hwang, J., & Park, N. (2022, June). Graph neural controlled differential equations for traffic forecasting. In Proceedings of the AAAI conference on artificial intelligence (Vol. 36, No. 6, pp. 6367-6374).

[4] Jin, M., Zheng, Y., Li, Y. F., Chen, S., Yang, B., & Pan, S. (2022). Multivariate time series forecasting with dynamic graph neural odes. IEEE Transactions on Knowledge and Data Engineering, 35(9), 9168-9180.

**Questions:**

Based on the *weaknesses* above please focus on the following aspects:
1. **Experiments:** Explain choices of datasets and baselines or extend the experimental evaluation (W1). Conduct qualitative evaluations of graph-based causal representations (W2).
2. **Limitations:** Perform a comprehensive computational cost analysis comparing GNeuralFlow with simpler and more complex methods to highlight efficiency gains (W3).
3. **Contribution:** Please better position the presented contribution among relevant works (W4).

**Limitations:**

It would be more complete to experimentally showcase the computational limitations, demonstrating in practice the scalability of the proposed method for real-world datasets.

---

> ### Author Rebuttal · Authors · 2024-08-07
>
> **RE: additional experiments (Q1)**
>
> Thanks for your comment.
> Here we provide additional baselines. Specifically, we provide the comparison with GRU-D (Che et al 2016), NRI (Kipf et al., 2018), and dNRI (Graber and Schwing 2020) on the synthetic datasets.
>
> We observe that our GNF-gru can improve over both Neural-flow with GRU and GRU-D. While GNF-gru is outperformed by NRI and dNRI on most datasets, our GNR-resnet achieves the best performance overall.
>
> | model             | Sink            | Triangle        | Sawtooth        | Square          |
> | ----------------- | --------------- | --------------- | --------------- | --------------- |
> | NRI               | 5.25(±0.02)     | 3.96(±0.16)     | 4.99(±0.12)     | 9.39 (±0.45)    |
> | dNRI              | 5.40(±0.04)     | 3.39(±0.09)     | 4.97(±0.21)     | 9.78(±0.21)     |
> | NF-GRU            | 10.9(±0.43)     | 10.3(±0.45)     | 16.1(±0.41)     | 17.2(±0.51)     |
> | GRU-D             | 12.31(±0.23)    | 11.25(±0.32)    | 17.55(±0.53)    | 18.73(±0.31)    |
> | GNF-resnet (ours) | **3.95(±0.32)** | **2.32(±0.11)** | **3.84(±0.06)** | **8.24(±0.64)** |
> | GNF-gru (ours)    | 6.83(±0.23)     | 5.41(±0.23)     | 5.11(±0.13)     | 9.14(±0.61)     |
>
>
>
>
> **RE: computational cost (Q2)**
>
> Thanks for the feedback. We present the training cost as the mean training time for one epoch (in seconds) over synthetic datasets. As expected, our graph neural flow (GNF) models are more expensive than the neural flow corresponding ones (because of the additional modeling of the graph). However, the GNF are still cheaper than the Neural ODEs (because we do not need a numerical solver).
>
>
> | Model               | Sink  | Triangle | Sawtooth | Square |
> | ------------------- | ----- | -------- | -------- | ------ |
> | Neural-ODE          | 1.529 | 1.527    | 1.742    | 2.206  |
> | NF-resnet           | 1.022 | 1.013    | 1.021    | 1.020  |
> | NF-coupling         | 0.136 | 0.137    | 0.136    | 0.133  |
> | NF-gru              | 0.251 | 0.249    | 0.247    | 0.247  |
> | GNF-resnet (ours)   | 1.521 | 1.521    | 1.534    | 1.533  |
> | GNF-coupling (ours) | 1.215 | 1.214    | 1.212    | 1.213  |
> | GNF-gru (ours)      | 0.275 | 0.283    | 0.286    | 0.284  |
>
>
> **RE: position the presented contribution among relevant works (Q3)**
>
> Thanks for pointing out the related works from Choi et al (2022) and Jin et al (2022). We will add them to the main text of our paper. There are some differences with our method.
>
> First, these papers take the Neural ODE approach, whereas our method follows neural flows. While both aim at solving an unknown ODE, neural flows are computationally more economical than Neural ODE, because they do not require the repeated calls of ODESolve. It is unclear if the referenced papers can straightforwardly swap Neural ODE with neural flows for this benefit, without changing other components of their models.
>
> Secondly, these papers consider time-varying graphs. Here, we are not to argue if a time-varying graph or a constant graph is superior; rather, they come from different modeling beliefs. Time-varying graph modeling typically constructs the graph from data (e.g., an affinity graph of given feature vectors at a time, or a co-occurrence graph of observations within a sliding time window) or parameterizes the graph based on node embeddings; whereas DAG structure learning treats the graph as a free parameter to learn. Our method models a constant graph that is assumed to generate the data over time.
>
>
> **References:**
>
> Choi et al (2022). Graph neural controlled differential equations for traffic forecasting. AAAI.
>
> Jin et al (2022). Multivariate time series forecasting with dynamic graph neural odes. IEEE Transactions on Knowledge and Data Engineering.
>
> Che et al (2016). Recurrent Neural Networks for Multivariate Time Series with Missing Values.
>
> Kipf et al. (2018). Neural Relational Inference for Interacting Systems, ICML.
>
> Graber and Schwing. (2020). Dynamic Neural Relational Inference, CVPR
>
>
> **Final comment**
>
> Thank you for the feedback. Please consider raising the score if we have addressed your concerns.

---

> > ### Comment · Reviewer_3M7x · 2024-08-09
> >
> > Thank you for the replies, which have partially addressed the issues raised by my side (e.g., baselines and computational costs). Considering other reviewers' comments as well as the impact of the contribution (when it comes to mathematical background, performance improvements and complexity), I would prefer to maintain my rating.

---

### Official Review · Reviewer_4eJC · 2024-07-15

**Soundness:** 3
**Presentation:** 3
**Contribution:** 2
**Rating:** 5
**Confidence:** 3

**Summary:**

This paper addresses the problem of multivariate time series prediction with feature interactions. It proposes learning a Directed Acyclic Graph (DAG) to model interactions, encoded by a Graph Neural Network (GNN), and using a neural flow to model the dynamics. The experiments demonstrate improvements over graph-free models and models with given graphs obtained from covariance.

**Strengths:**

The combination of graph learning and neural flow, along with DAG-conditioned optimization, is original. Since flows are popular in generative modeling, the idea of converting them to the graph setting seems promising. The paper also learns graph parameters using the matrix A to learn (or relearn the DAG structure), which can be useful in cases where the graph is not previously known with certainty.

**Weaknesses:**

Weaknesses:
Flow matching inherently has the idea of using a linear interpolant (straight line) to encode optimal transport paths in time series data. In spaces where this is not true, or where the optimal transport paths are perhaps in a latent dimension the assumptions are violated. This is why Flow matching has been more useful for generative modeling than time series. This is not noticed by the authors and the restrictive experiments they perform are insufficient to test this.

I don't see the need for a DAG here. See the RITINI model [Bhaskar et al, LOG 2023], when there is time series data there can be feedback loops in the variables rather than DAG-style causality. I would appreciate trying to loosen this requirement, which would enable cyclic dependencies.

The set of comparisons featured here is rather limited. Here are several works that could be compared against:

Kipf et al., Neural Relational Inference for Interacting Systems, ICML, 2018.
Graber and Schwing. Dynamic Neural Relational Inference, CVPR, 2020 (dynamic graph inference for discrete-time systems)
Nishikawa-Toomey et al., Bayesian learning of Causal Structure and Mechanisms with GFlowNets and Variational Bayes (jointly learn the DAG and the parameters of a linear-Gaussian causal model)
Deleu et al., Joint Bayesian Inference of Graphical Structure and Parameters with a Single Generative Flow Network, NeurIPS, 2023 (jointly learn the posterior over the structure of a Bayesian Network, and also the parameters of its conditional probability distributions)
Smith and Zhou, Coordinated Multi-Neighborhood Learning on a Directed Acyclic Graph, arXiv:2405.15358 (constraint-based approach that exploits conditional independence relationships between variables to infer the underlying causal model)
Hiremath et al. Hybrid Global Causal Discovery with Local Search, arXiv:2405.14496 (global causal discovery by learning local causal substructures using topological sorting and pruning of spurious edges)
Bhaskar et al. Inferring dynamic regulatory interaction graphs from time series data with perturbations, LOG 2023

**Questions:**

For the experiments on synthetic datasets (Figure 1), how are the graphs for graph ODEs obtained? Are all models using the ground truth graph? If so, how is the conclusion that “DAG is better than other graph structures at modeling dependencies” (line 252) drawn?

Does the performance gain over existing graph ODE models come from the learnable graph? Is there an ablation study for that?

Is the model learning a constant graph independent of time? How does it perform when there is a significant change in data dependence?

**Limitations:**

The authors have adequately addressed the limitations.

---

> ### Author Rebuttal · Authors · 2024-08-07
>
> **RE: flow matching (W1)**
>
> Our effort consists in extending Neural Flows (Bilos et al 2021), to integrate additional interdependency information in the form of a learned graph. Note that (Bilos et al 2021)  was proposed in the context of time series, differently from Flow Matching. We are not proposing a flow-matching approach.
>
>
> **RE: no need for a DAG here (W2)**
>
> Thanks for pointing out this perspective. Our view on the matter is the following.
>
> A DAG provides a clear and interpretable representation of causal relationships between variables. In the context of time series data from multiple sensors, understanding these causal relationships can help identify how changes in one sensor's readings might influence others. This is crucial in medical monitoring, where understanding causality can aid in diagnosing conditions or understanding physiological responses.
>
> In addition, DAG structure learning is a long-lasting and challenging problem in probabilistic graphical models (in particular, Bayesian networks). The structure of a Bayesian network takes the form of a DAG and plays a vital role in causal inference (Pearl, 1985). Learning the DAG structure is a combinatorial problem; it is not only theoretically intimidating (NP-hard; see Chickering et al. (2004)) but also practically challenging even when approximate solutions are sought.
>
> **References:**
>
> Bilos et al (2021).  Neural flows: Efficient alternative to neural ODEs. Neurips.
>
> Pearl (1985). Bayesian networks: A model of self-activated memory for evidential reasoning. Proceedings of the 7th Conference of the Cognitive Science Society.
>
> Chickering et al. (2004). Large-sample learning of Bayesian networks is NP-hard. JMLR.
>
>
>
> **RE: related works (W3)**
>
> Thanks for providing these related works, we will place them appropriately in the main text.
>
> Here we add the comparison with two of the requested baselines. Specifically, we provide a comparison with NRI (Kipf et al., 2018) and dNRI (Graber and Schwing 2020) methods on the synthetic datasets.
>
>
> | model      | Sink            | Triangle        | Sawtooth        | Square          |
> | ---------- | --------------- | --------------- | --------------- | --------------- |
> | NRI        | 5.25(±0.02)     | 3.96(±0.16)     | 4.99(±0.12)     | 9.39 (±0.45)    |
> | dNRI       | 5.40(±0.04)     | 3.39(±0.09)     | 4.97(±0.21)     | 9.78(±0.21)     |
> | GNF-resnet | **3.95(±0.32)** | **2.32(±0.11)** | **3.84(±0.06)** | **8.24(±0.64)** |
>
> We notice that three of the requested comparisons were posted after the conference deadline, namely, Nishikawa-Toomey, (2024), Smith and Zhou (2024), and Hiremath et al. (2024).
>
>
>
> **References:**
>
> Kipf et al. (2018). Neural Relational Inference for Interacting Systems, ICML.
>
> Graber and Schwing. (2020). Dynamic Neural Relational Inference, CVPR
>
> Hiremath et al. (2024). Hybrid Global Causal Discovery with Local Search, arXiv
>
> Smith and Zhou (2024). Coordinated Multi-Neighborhood Learning on a Directed Acyclic Graph. Arxiv.
>
> Nishikawa-Toomey et al. (2024). Bayesian learning of Causal Structure and Mechanisms with GFlowNets and Variational Bayes. Arxiv.
>
>
> **RE: how are the graphs for graph ODEs obtained, and why DAG is better than other graph structures at modeling dependencies (Q1)**
>
> Thanks for raising this point. The graph ODEs are given the ground truth graph in input for the synthetic data, while they are given the covariance matrix in the real-world datasets.
>
> Empirically, we find that when we employ our DAG method, we achieve improvements over graph-ODEs, even while they are given the ground truth graph.
>
>
> **RE: performance gain over existing graph ODE (Q2)**
>
> Thanks for raising this point. The performance gain is due to the DAG component. For example, in Figure 2, we show that a NeuralFlow model without graph dependencies performs worse than both GNeuralFlow with a learned graph and with the ground truth graph.
>
>
> **RE: learning a constant graph independent of time (Q3)**
>
> While we learn a constant graph, and certain datasets may not work under such an assumption, in the experiments we achieved success also in real-life datasets. Moreover, our framework can support learning a dynamical graph by parameterizing it with time-dependent node embeddings.

---

> > ### Comment · Reviewer_4eJC · 2024-08-13
> > **Clarified neural flows**
> >
> > Thanks for the clarification of the neural flows approach, in other words I see that you directly model the integral curve.  I think this is also interesting for a graph. I also acknowledge the effort of adding the baselines. The idea that you would use a DAG because others do is not a great justification, we know medicine is filled with circularly dependent variables. Therefore I will raise my score slightly to 5.

---

### Official Review · Reviewer_ufSP · 2024-07-19

**Soundness:** 3
**Presentation:** 3
**Contribution:** 3
**Rating:** 6
**Confidence:** 4

**Summary:**

The paper proposes a continuous-time model to discover the causal structure from irregular multivariate time series data. The idea is to introduce the DAG (directed acyclic graph) to model the interaction of different time series at the same time step in the vector field that generates the multivariate time series（that is, the $n$ trajectories). Instead of learning the vector field, the paper proposes to learn the solution of the ODE system directly with neural flows. The structural interaction is implemented with graph convolution neural networks and three particular parameterization methods are proposed for the solution function $F$ to guarantee its invertibleness. The experiments on both synthetic and real-world irregular time series datasets are conducted to assess the efficacy of the proposed method for both prediction and classification tasks.

**Strengths:**

* The proposed method is well motivated and clearly illustrated, and the presentation is easy to follow. The reviewer appreciates the way to establish the solution framework presented in Section 4.1, which is very well explained.
* It is interesting to introduce the DAG structure learning to irregular time series modeling and the proposed method seems reasonable to the reviewer.
* The proposed method achieves better results than the existing method.

**Weaknesses:**

* In Section 3.2, it states "(i) the vector fields $\mathbf{B}\mathbf{x}^1, \ldots, \mathbf{B}\mathbf{x}^n$ follow the same conditional dependence structure governed by $\mathbf{A}$",  but according to Eq. (2), the vector fields seems to be $\mathbf{B}\mathbf{x}^j - \sum_{i=1}^{j-1}a_{ij} \mathbf{Bx}^{i}$ for $j=1,\ldots,n$, so I would suggest changing the description to "the vector fields that generate the $n$ trajectories follow the same conditional dependence structure governed by $\mathbf{A}$" to avoid confusion.
* To learn the DAG matrix, the proposed method employs NOTEARS and DAG-GNN. How do these DAG learning algorithms impact the model performance?

* The proposed method (graph neural flows) is dubbed GNeuralFlow in the main text but is referred to as GNF in the tables of experiments. It is better to unify the abbreviation. Besides, no code is released.

**Questions:**

See Weaknesses.

**Limitations:**

No potential negative societal impact.

---

> ### Author Rebuttal · Authors · 2024-08-07
>
> **RE: ..I suggest changing the description to "the vector fields..**
>
> Thanks for the feedback. We will update the text.
>
>
> **RE: .. How do these DAG learning algorithms impact the model performance**
>
> Thanks for pointing out this part. We address it as follows.
>
> A limitation of the proposed model is that the number of parameters on the $A$ part grows quadratically with the number of time series (nodes). Such a scalability challenge is a common problem for DAG structure learning. While past research showcased the feasibility of learning a graph with a few hundred nodes (Yu et al 2019), going beyond is generally believed to require a new computational technique.
>
>
> **References**
>
> Yu et al 2019. DAG-GNN: DAG structure learning with graph neural networks. ICML
>
>
> **RE: unify the abbreviation GNF**
>
> GNF is commonly an abbreviation for Graph Normalizing Flow. We will change the abbreviation.
>
>
> **Final comment**
>
> Thank you for the comments. Please consider raising the score if we have addressed them appropriately.

---

> > ### Comment · Reviewer_ufSP · 2024-08-11
> >
> > Thanks for further clarification. I will keep my score.

---

### Decision · Program_Chairs · 2024-09-25

**Decision:**

Accept (poster)

**Comment:**

The proposed approach is found to be well motivated and well presented whereas the results are also found to be convincing. The approach is further considered both original and to have wide applications.

During the rebuttal phase, the authors included additional experimentation and comparison to baselines as well as clarified computational aspects of their modeling framework. This was appreciated by the reviewers and found to improve the paper. Consequently, the reviewers in general agreed that the manuscript is acceptable for publication. However, it was also discussed that the authors should include code for reproducibility.

The authors are thus encouraged to update their manuscript according to the additional experimentations and clarifications in their final version. Furthermore, they are also most strongly encouraged to include code in their final manuscript for reproducibility that unfortunately was not included in their original submission.